# Hemodynamic Effects of SGLT2 Inhibitors in Patients with and Without Diabetes Mellitus—A Narrative Review

**DOI:** 10.3390/healthcare12232464

**Published:** 2024-12-06

**Authors:** Roxana Brata, Andrei Vasile Pascalau, Ovidiu Fratila, Ioana Paul, Mihaela Mirela Muresan, Andreea Camarasan, Tiberia Ilias

**Affiliations:** 1Department of Medical Disciplines, Faculty of Medicine and Pharmacy, University of Oradea, P-ta 1 December 10, 410073 Oradea, Romania; roxana.gavrila@yahoo.com (R.B.); ovidiufr@yahoo.co.uk (O.F.); paul.ioanalarisa@gmail.com (I.P.); 2Department of Morphological Disciplines, Faculty of Medicine and Pharmacy, University of Oradea, P-ta 1 December 10, 410073 Oradea, Romania; elamuresan66@yahoo.com (M.M.M.); andreutza_2000@yahoo.com (A.C.)

**Keywords:** SGLT2 inhibitors, hypertension, chronic kidney disease, heart failure

## Abstract

**Background**: The current review aims to present the beneficial effects of SGLT2 inhibitors (dapagliflozin and empagliflozin) on several hemodynamic parameters such as blood pressure, filtration pressure at the level of the glomerular capillaries, and the improvement of the preload and afterload of heart muscle. In order to stop chronic kidney disease (CKD) from progressing, SGLT2 inhibitors have become an important disease-modifying treatment. **Materials and methods**: Recent clinical studies have shown the success of these drugs in treating heart failure, reducing the risk of cardiovascular events, hospitalization, and mortality. **Results**: The hemodynamic effects of SGLT2 inhibitors include a diuretic effect, due to reduced sodium reabsorption. Also, at this level, numerous studies have confirmed the beneficial effect of dapagliflozin in patients with chronic kidney disease, associated with a 44% reduced risk of progression in this pathology. SGLT2 inhibitors are associated with a reduction in blood pressure and weight loss, because of their diuretic effect, especially empagliflozin, which can explain the beneficial effects in patients with heart failure. In addition, mainly empagliflozin reduces stiffness and arterial resistance. **Conclusions**: Although the exact mechanism of action is unknown, SGLT2 inhibitors reduce the interstitial volume by blocking the tubular reabsorption of glucose. This leads to reduced blood pressure and enhanced endothelial function. Consequently, there have been improvements in hospitalization and fatality rates. Because of their beneficial effects, these medications have been guidelines for managing heart failure and chronic kidney disease.

## 1. Introduction

Sodium–glucose cotransporter 2 (SGLT2) inhibitors are novel medications that act by inhibiting the co-transport of sodium and glucose at the level of the proximal convoluted tubule of the kidney. Therefore, these medications produce glycosuria and natriuresis [1]. The main representatives of the class are dapagliflozin, empagliflozin, and canagliflozin, but sotagliflozin and ertugliflozin have also been introduced to the market. They were first used in the treatment of type 2 diabetes mellitus (DM), and the beneficial hemodynamic effects were soon observed and they were introduced into heart failure and chronic kidney disease (CKD) treatment in both diabetic and non-diabetic patients [2]. Globally, a high number of people suffer from heart failure and CKD. Two very large trials demonstrated the beneficial effects of dapagliflozin on heart failure and CKD in patients with and without diabetes mellitus [3,4].

In addition to standard guideline-directed medical therapy, 4744 patients with symptomatic heart failure and a reduced ejection fraction (≤40%) were given either dapagliflozin 10 mg or a placebo in the Dapagliflozin and Prevention of Adverse Outcomes in Heart Failure (DAPA-HF) trial. In this trial, patients receiving dapagliflozin compared with the placebo experienced a lower rate of the primary composite outcomes, which included worsening heart failure (unplanned hospitalization or an urgent visit for heart failure) or death from cardiovascular causes (16.3% vs. 21.2%; *p* < 0.001) [3].

In another very large trial, 4304 individuals with or without diabetes mellitus, who had an estimated glomerular filtration rate (GFR) of 25 to 75 mL per minute per 1.73 m^2^ of body surface area, were randomly assigned to receive either a placebo or dapagliflozin (10 mg once day). A composite of end-stage kidney disease, cardiovascular or renal mortality, or a persistent reduction in the estimated GFR of at least 50% was the primary clinical outcome. Regardless of whether the patients with CKD had diabetes or not, dapagliflozin treatment significantly decreased the risk of a composite of end-stage kidney disease and mortality from renal or cardiovascular causes, and resulted in a persistent reduction in the estimated GFR of at least 50% [4].

Starting from the results of these two well-documented trials, which suggest that SGLT2 inhibitors exhibit a positive role in the natural evolution of heart failure and CKD, we set out to clarify which is the potential pathogenetic mechanism that leads to the observed treatment effects.

The current review aims to clarify the hemodynamic mechanisms that act as a basis for the beneficial effects of SGLT2 inhibitors in heart failure and CKD.

## 2. Materials and Methods

We conducted a thorough search of the PubMed, Scopus, and Web of Science databases for review articles or original articles. The main keywords searched were: “hemodynamic effects” and “SGLT2 inhibitors”. The search was limited to English language and human studies, or studies conducted on animal models on the hemodynamic effects of SGLT2 inhibitors. The search included articles published between 2014 and 2024. After the primary selection, we used the following inclusion criteria: articles that discussed specified hemodynamic effects (renal, cardiovascular, effects on blood pressure, and effects on glomerular function, etc.), or large clinical trials that addressed the beneficial renal or cardiovascular effects (Table 1). The exclusion criteria were articles lacking full text, articles of low-quality or irrelevant research. The steps of the search process are illustrated in Figure 1.

## 3. Discussions—Hemodynamic Effects of SGLT2 Inhibitors

### 3.1. Natriuresis and Diuresis

Due to reduced sodium reabsorption at the level of the proximal convoluted tubule, SGLT2 inhibitors have a diuretic effect. Increased glycosuria also contributes to this effect, with it being well known that glucose is an osmotic active substance [5]. However, in clinical practice, it has been observed that when compared to loop diuretics such as furosemide, SGLT2 inhibitors produced a reduction in interstitial volume, as well as a lesser reduction in intravascular volume; therefore, SGLT2 inhibitors have an advantage over loop diuretics, because they do not produce the reflex neurohormonal activation with tachycardia that is often encountered in loop diuretic treatment [6]. There were no changes in natriuresis or plasma volume in the study among participants who received dapagliflozin and had urine collected 24 h before treatment, after acute dosing, after two weeks of treatment, and three days after treatment cessation [7]. Although the proximal tubule reabsorbs most of the sodium, the precise contribution of SGLT2 transporters to total sodium reabsorption remains unknown. This phenomenon may be particularly relevant in people with type 2 diabetes mellitus, in whom there is increased glucose flux through the SGLT2 transporters, owing to chronic tubular hyperglycaemia. In the proximal tubule, SGLT2 transporters also functionally interact with Na^+^/H^+^ exchanger isoform 3 (NHE3). The predominant epithelial isoform of NHE3 is found in apical membranes of renal and intestinal epithelial cells, where it contributes to NaCl reabsorption [8].

To maintain a neutral sodium balance, the kidneys quickly adjust to the initial natriuresis by balancing sodium excretion with sodium reabsorption. Because of compensatory sodium reabsorption at more distal tubular segments, sodium excretion is typically not affected by prolonged treatment with SGLT2 inhibitors [9,10].

Uncertainty surrounds the location of the additional sodium absorption in the distal tubular segments. Since the macula densa is thought to detect higher sodium and chloride concentrations, any increase in renal sodium absorption caused by SGLT2 inhibition is most likely distal to the macula densa. This is because SGLT2 inhibition causes a decrease in the estimated glomerular filtration rate through tubule-glomerular feedback [8].

These drug classes exhibit significant differences in their effects on potassium, uric acid, glucose, renal hemodynamics, and markers of the renin–angiotensin–aldosterone system, possibly due to their distinct sites of action within the tubular system. Studies are conflicting regarding the diuretic effects of SGLT2 inhibitors; some studies have reported that because of their diuretic/natriuretic effects, they lead to slightly raised renin levels [11,12]. Our model predicts that the natriuretic and diuretic effects of SGLT2 inhibition are reduced but maintained in normoglycemia compared to diabetes, and with renal impairment compared to normal renal function, which is consistent with recent simulations by Layton and Vallon [11]. Furthermore, we hypothesized that decreases in glomerular pressure and SNGFR were almost independent of renal function and greater in diabetes than in normoglycemia. However, compared to intact renal function, the overall GFR decline was less pronounced in cases of renal impairment. This is because hyperfiltration occurs in diabetic nephrons, and the degree of hyperfiltration in the remaining nephrons increases as nephrons are destroyed. Because of both hyperfiltration and hyperglycemia, the remaining nephrons in diabetics with renal impairment are filtering more glucose and Na^+^ [11].

### 3.2. Reduction of Filtration Pressure at the Glomerular Level

Also, because of the reduction in sodium reabsorption, more sodium ions are delivered to the level of the macula densa. If a large quantity of sodium ions is reabsorbed in the proximal convoluted tubule, a small amount of sodium ions arrive at the level of the macula densa, and this initiates the tubule-glomerular feedback mechanism, which produces the vasodilatation of the afferent arteriole and increased glomerular pressure, a pathophysiological mechanism responsible for the development of glomerular injury. This mechanism is responsible for the development of chronic kidney disease characterized by microalbuminuria and the loss of kidney function [13]. Numerous studies conhave firmed the beneficial effects of SGLT2 inhibitors in CKD such as the study Dapagliflozin in Patients with chronic kidney disease (DAPA-CKD), where dapagliflozin administered in a dose of 10 mg/day in patients with or without diabetes mellitus was associated with a 44% reduced risk of progression to end stage kidney disease or death of renal causes [4].

An increase in the intracellular Na^+^ concentration at the level of the macula densa results from the inhibition of SGLT2, which is normally upregulated in diabetic kidneys. This leaves a higher-than-normal concentration of Na^+^ in the proximal tubule lumen, and after passing through the loop of Henle, the macula cells’ uptake of this Na^+^ exceeds the capacity of their Na^+^-K^+^-ATPase in the basolateral membrane [14]. Macula densa cells increase in volume and release ATP that is converted to adenosine. Therefore, SGLT2 inhibitors lead to an increased production of adenosine at the level of the macula densa. In cortical nephrons, adenosine A1 receptor-mediated constriction of the afferent arterioles may be predominant, whereas juxtamedullary or deep cortical nephrons may exhibit more pronounced adenosine A2 receptor-mediated vasodilation [14]. The prolonged stimulation of tubule-glomerular feedback by SGLT2 inhibitors could be explained by several possible processes, such as the suppression of NHE3 or the lack of Na^+^-K^+^-2Cl^−^ cotransporter (NKCC) alterations. SGLT2 inhibitors appear to block the sodium chloride cotransporter (NCC), epithelial sodium channel (ENaC), and aquaporin-2 (AQP2) in the distal nephron via unknown mechanisms that may involve interactions between adenosine receptors. Continuous natriuresis and aquaresis may be facilitated by these interactions [15].

### 3.3. Blood Pressure Reduction

SGLT2 inhibitors are associated with a reduction in blood pressure mainly because of their diuretic effect; however, the moderate weight loss that SGLT2 inhibitors produce because of increased urine glucose elimination could also be a cause of blood pressure reduction [16]. A large meta-analysis that included 2381 participants treated with SGLT2 inhibitors produced an average reduction of 3.62/1.70 mmHg [17]. Another meta-analysis that included 22,528 patients from 43 randomized control trials demonstrated a significant reduction in systolic blood pressure (−2.46 mmHg (95% CI −2.86 to −2.06)) and diastolic blood pressure (−1.46 mmHg (95% CI −1.82 to −1.09)) with SGLT2 inhibitors compared to a placebo or other agents [18].

The effects of SGLT2 inhibitors on 24-h ambulatory blood pressure was examined through a meta-analysis of randomized, double-blind, placebo-controlled trials. We included every study reporting 24-h ambulatory blood pressure data that was published before 17 August 2016. A random-effects model was used to calculate the mean differences in 24-h blood pressure, blood pressure during the day, and blood pressure at night. Significant reductions in 24-h ambulatory systolic and diastolic blood pressure are observed with SGLT2 inhibitors: −3.76 mm Hg (95% CI, −4.23 to −2.34; I2 = 0.99) and −1.83 mm Hg (95% CI, −2.35 to −1.31; I2 = 0.76), respectively. Significant systolic and diastolic blood pressure reductions were also observed during the day and at night [19].

Following the administration of SGLT2 inhibitors, systolic blood pressure (SBP) decreased for up to six months. An increase in urine sodium excretion was correlated with a decrease in SBP from baseline at six months, rather than an increase in urinary glucose excretion. This suggests that plasma volume reduction from natriuresis mediated by SGLT inhibition, rather than osmotic diuresis, could be the primary cause of the BP-lowering effect of SGLT2 inhibitors at six months. Six months after starting to take SGLT2 inhibitors, diastolic blood pressure (DBP) was significantly reduced for the first time. After a month, there was a noticeable reduction in body weight, which persisted for another six months [20].

A reduction in blood pressure (BP) may be partially responsible for the better cardiovascular outcomes. The administration of SLGT2 inhibitors reduces systolic blood pressure by about 4 mmHg [21]. Although the exact mechanisms behind the BP-lowering effect are still unknown, a persistently decreased plasma volume may be a factor [22,23]. It has been proposed that sympathetic nervous system (SNS) activity is also decreased, because decreases in plasma volume and (arterial) blood pressure happen without an increase in heart rate (HR). It has also been suggested that improvements in endothelial function and arterial stiffness may play a role in long-term BP reduction [24,25].

### 3.4. Improvement in Preload and Afterload

Because of glycosuria, natriuresis, osmotic effect and the preload of the left ventricle are improved in treatment with SGLT2 inhibitors. Magnetic resonance imaging (MRI) data demonstrated reduced end-diastolic volumes in patients treated with SGLT2 inhibitors [26]. Much evidence has appeared suggesting that SGLT2 inhibitors improve arterial stiffness, which decreases afterload.

Significantly for HF, preload may be decreased by SGLT2 inhibition-mediated effects on effective circulating volume contraction, which would lower ventricular filling pressure. Overall, it would be anticipated that SGLT2 inhibition would lower cardiac preload, due to its effects on natriuresis and the corresponding plasma volume contraction. Afterload reductions could happen because of lowering blood pressure and arterial stiffness, which would improve subendocardial blood flow [27].

In addition to normal to slightly elevated left ventricular volumes and systolic blood pressures, oxidative stress and inflammation can cause fibrosis of the underlying myocardium, expanded epicardial adipose tissue mass, microvascular endothelial dysfunction, and increased arterial wall stiffness, all of which can lead to preserved ejection fraction (HFpEF) [28]. Research has indicated that SGLT2 inhibitors may improve microcirculatory dysfunction, lower systemic blood pressure, and lessen oxidative stress and inflammatory responses in HFpEF [29,30]. SGLT2 inhibitors have also been linked to decreased left ventricular mass and excessive diastolic tension, which improves cardiac preload, according to several studies [31]. Additionally, because they interfere with metabolic pathways, SGLT2 inhibitors may partially alleviate the symptoms of HFpEF [32].

SGLT2 inhibitors cause a state of metabolism known as ketogenic metabolism, which boosts the efficiency of the myocardium and the kidneys by using energy-efficient ketones, instead of less efficient fatty acid and glucose oxidation to produce myocardial energy [33].

Empagliflozin treatment has been associated with a decrease in central systolic blood pressure, central pulse pressure, and reflected wave amplitude [34]. All these data can explain the beneficial effects observed in patients with heart failure in large, randomized controlled trials when they were administered with SGLT2 inhibitors. In the DAPA-HF trial, a 30% reduction in the risk of worsening heart failure has been observed in patients with or without diabetes mellitus when administered dapagliflozin compared to a placebo. This study included patients with a reduced ejection fraction (HFrEF) of less than 40% [3]. These beneficial effects have also been observed in patients with a preserved ejection fraction (HFpEF) of about 60%, where dapagliflozin administration in a dose of 10 mg/day reduced the symptom severity of these patients, and were measured with the Kansas City Cardiomyopathy Questionnaire Clinical Summary Score (KCCQ-CS) and the physical limitations score [35]. These significant results have led to the inclusion of SGLT2 inhibitors in the European Society of Cardiology’s 2021 guidelines for heart failure treatment, as an add-on to classic therapy with angiotensin-converting enzyme inhibitors/angiotensin–receptor–neprilysin inhibitors (ACEI/ARNI), beta-blockers or mineralocorticoid receptor antagonists (MRA) [36]. The beneficial hemodynamic effect of SGLT2 inhibitors is exposed in Figure 2.

### 3.5. Summary of the Beneficial Hemodynamic Effects Obtained in Large Clinical Trials

Whereas the inclusions of findings from trials like the DAPA-CKD and DAPA-HF strengthen the argument for the therapeutic benefits of SGLT2 inhibitors in CKD (chronic kidney disease) and HF (heart failure), some meta-analysis and randomized controlled trials are cited, providing a robust evidence base for clinical effects (Table 2). Those results, where blood pressure reduction and improvements in HF symptoms represent the main purpose in this section, can be completed with detailed discussions of renal mechanisms, such as the interaction between the Na^+^/H^+^ exchanger isoform 3 (NHE3) and SGLT2 inhibitors, macula densa signalling, and adenosine receptor-mediated effects. All of them are well presented in the current review, including multiple hemodynamic effects of SGLT2 inhibitors, especially their natriuretic/diuretic effects.

**Table 2 healthcare-12-02464-t002:** SGLT2 renal and cardiovascular outcome studies.

Study Name	Main Outcome	Included Patients	Reference
DAPA-HF	Dapagliflozin reduced cardiovascular death or worsening heart failure events compared to the placebo.	4744 patients with HFrEF (LVEF ≤ 40%), with or without type 2 diabetes.	[3]
EMPEROR-Reduced	Empagliflozin reduced the risk of the composite of cardiovascular death or heart failure hospitalization.	3730 patients with HFrEF (LVEF ≤ 40%), with or without diabetes.	[37]
EMPEROR-Preserved	Empagliflozin reduced the combined risk of cardiovascular death or heart failure hospitalization in HFpEF patients.	5988 patients with HFpEF (LVEF > 40%), with or without diabetes.	[38]
DAPA-CKD	Dapagliflozin reduced the risk of the composite outcome of sustained decline in eGFR, end-stage kidney disease, or renal/cardiovascular death.	4304 patients with chronic kidney disease (eGFR 25–75 mL/min/1.73 m^2^), with or without type 2 diabetes.	[4]
DECLARE-TIMI 58	Dapagliflozin significantly reduced the risk of heart failure hospitalization and renal outcomes.	17,160 patients with type 2 diabetes, with or without prior cardiovascular disease.	[39]
CANVAS Program	Canagliflozin reduced the risk of major adverse cardiovascular events (MACE) and provided renal protection, but increased risk of amputation.	10,142 patients with type 2 diabetes at high cardiovascular risk.	[40]
CREDENCE	Canagliflozin significantly reduced the risk of renal failure, cardiovascular death, and heart failure hospitalization in CKD patients.	4401 patients with type 2 diabetes, CKD (eGFR 30–90 mL/min/1.73 m^2^) and albuminuria.	[41]
VERTIS-CV	Ertugliflozin did not significantly reduce MACE but showed benefits in heart failure hospitalization reduction.	8246 patients with type 2 diabetes and established cardiovascular disease.	[42]
SOLOIST-WHF	Sotagliflozin significantly reduced cardiovascular death, heart failure hospitalization, and urgent visits for heart failure.	1222 patients with type 2 diabetes and recent worsening heart failure.	[43]
SCORED	Sotagliflozin reduced the risk of cardiovascular death, heart failure hospitalization, and urgent heart failure visits in CKD patients.	10,584 patients with type 2 diabetes and CKD (eGFR 25–60 mL/min/1.73 m^2^).	[44]

The table above summarizes the beneficial effects of SGLT2 inhibitors administration in patients with heart failure, with various grades of ejection fraction or CKD obtained in large clinical trials demonstrating that the hemodynamic effects discussed above reflect an improved clinical outcome for the patient.

Table 2 shows the main drugs belonging to the class called SGLT2 inhibitors and their main effects on morbidity and mortality, as follows.

A representative study ranked the dapagliflozin effect. Whether or not diabetes mellitus was present, patients with heart failure and reduced ejection fraction who received this drug had a lower risk of death or worsening heart failure than those who received a placebo [3].

The primary composite outcome of a sustained GFR of at least 50%, end-stage kidney disease, or death from renal or cardiovascular causes, was less likely to occur in participants with chronic kidney disease, with or without type 2 diabetes, who were randomly assigned to receive dapagliflozin, than in those who were assigned to receive a placebo [4].

Another study based on dapagliflozin SGLT2 inhibitors presented a large group of individuals with or at risk of atherosclerotic cardiovascular disease, and assessed the drug’s effects on cardiovascular and renal outcomes. Dapagliflozin treatment did not result in a higher or lower rate of MACE (major adverse cardiovascular events) than the placebo in patients with type 2 diabetes who had or were at risk of atherosclerotic cardiovascular disease; however, it did result in a lower rate of cardiovascular death or hospitalization for heart failure, which is indicative of a lower rate of heart failure hospitalization [39].

The EMPEROR study presented that the people from the empagliflozin group who were getting the prescribed treatment for heart failure were less likely to die from cardiovascular causes or be admitted to the hospital due to heart failure than those in the placebo group [37].

Empagliflozin-induced SGLT2 inhibitors reduced the composite risk of cardiovascular death or heart failure hospitalization by 21% in patients with heart failure and a preserved ejection fraction. This was primarily due to a 29% decreased risk of heart failure hospitalization. Across all predefined categories, including patients with or without diabetes, the effects on the incidence of primary outcome events were largely consistent [38].

Another specific drug with beneficial effects on MACE is canagliflozin. Two out of forty-seven selected articles present the particular impact of a sodium–glucose cotransporter 2 inhibitor called canagliflozin in clinical practice, and how this drug lowers blood pressure, body weight, albuminuria, and glycemia in diabetics [40,41]. Canagliflozin may also reduce the risk of serious cardiovascular and renal complications, and also death [40]. In conclusion, according to preliminary trials, people with type 2 diabetes may benefit from improved renal outcomes while using SGLT2 inhibitors [41].

Ertugliflozin was not worse than a placebo in patients with type 2 diabetes mellitus and atherosclerotic cardiovascular disease in terms of significant adverse cardiovascular events [42].

Sotagliflozin therapy, whether started prior to or soon after hospital discharge, reduced the overall number of deaths from cardiovascular causes, urgent visits for heart failure in the trial involving patients with diabetes, and a recent episode of acute decompensated heart failure compared to a placebo [43,44].

### 3.6. Other Beneficial Hemodynamic Effects

By reducing endothelial cell activation, promoting direct vasorelaxation, and addressing endothelial dysfunction or the expression of pro-atherogenic cells and molecules, SGLT2 inhibitors reduce arterial stiffness and vascular resistance [25]. Empagliflozin works at the level of the mitochondrial metabolism, as well as mediating the interaction between cardiac microvascular endothelial cells and cardiac myocytes, to reduce microvascular dysfunction [45]. Empagliflozin decreases arterial stiffness in aged mice by upregulating pathways involved in the synthesis of reactive oxygen species and increasing nitric oxide synthase (eNOS) activation, according to a similar study [46]. Endothelial cells are directly affected by SGLT2i-induced endothelial protection mechanisms, which reduce oxidative stress and increase the availability of NO and endothelial cell survival. Simultaneously, systemic in vivo models of heart failure or myocardial injury show a consistent reduction in adhesion molecules and chemokine release, demonstrating the anti-inflammatory properties of SGLT2. Although a large body of preclinical research supports SGLT2 is’ protective function on the endothelium, it is difficult to pinpoint which precise molecular targets SGLT2 is directly affect. This restriction is consistent with the lack of a direct demonstration of the SGLT2 transporter’s expression in endothelial cells or blood vessels [47].

A relevant comparison between SGLT2 inhibitors and other drugs can be revealed in the next section.

An important study showed that urine osmolarity and glucose excretion increased from the first day after dapagliflozin administration, compared with individual treatment with bumetanide. In the same study, the daily excretion of uric acid and fluid was greater after the combined administration of these drugs than after bumetanide alone [12]. Regarding serum urate levels, we can observe an important effect of dapagliflozin, which, when used alone, decreases these levels by 36%, in contrast to the use of bumetanide, which increases this serum level by 3% [12].

Another significant study showed the importance of using these SGLT2 inhibitors in combination with diuretic medication.

A relevant example refers to the protective cardiovascular effects of SGLT2 inhibitors that surpassed even chlorthalidone [17].

## 4. Conclusions

We identified several mechanisms by which SGLT2 inhibitors improved hemodynamic function: osmotic effect, natriuresis, reduction of kidney filtration pressure, improvement in heart preload and afterload, and reduction of blood pressure. Although some mechanisms are clear, others, such as the diuretic or natriuretic effect, remain to be clarified in further experimental and clinical research. We believe that these beneficial effects of SGLT2 inhibitors should be known by all clinicians involved in the treatment and prevention of heart disease and kidney disease.

## Figures and Tables

**Figure 1 healthcare-12-02464-f001:**
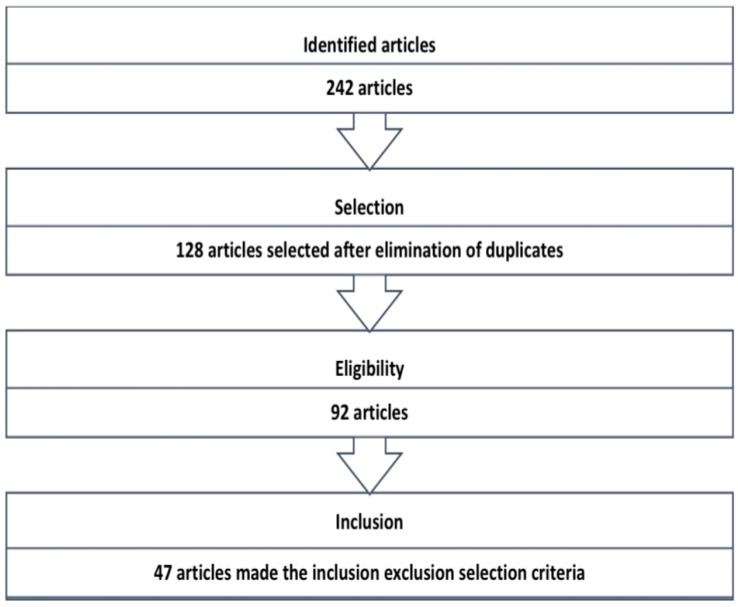
Flow-chart of article selection for study inclusion.

**Figure 2 healthcare-12-02464-f002:**
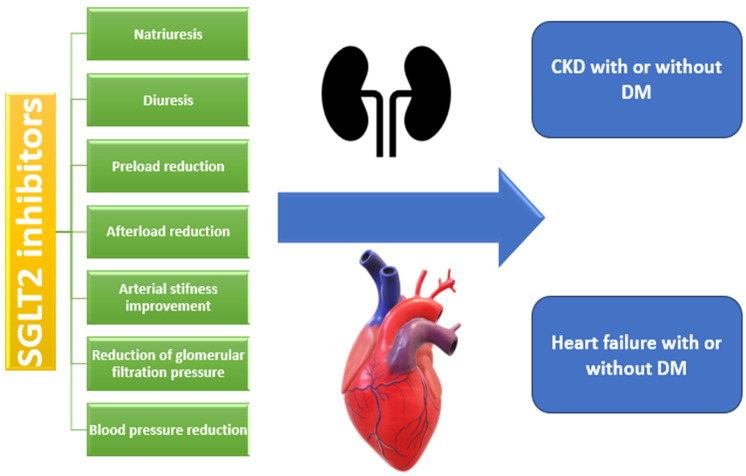
Beneficial hemodynamic effects of SGLT2 inhibitors (F developed with Office Developer Tools) based on data in the article. The upper part of the image, represented by the kidney diagram, suggests that the patients with CKD (chronic kidney disease) treated with SGLT2 inhibitors, regardless of the presence or absence of DM (Diabetes Mellitus), have obtained better results regarding glomerular filtration pressure, natriuresis, and diuresis. The lower image, represented by the heart diagram, suggests that the patients with heart failure treated with SGLT2 inhibitors, regardless of the presence or absence of DM (Diabetes Mellitus), have obtained an improvement in arterial stiffness, preload and afterload reduction, and blood pressure.

**Table 1 healthcare-12-02464-t001:** Inclusion/exclusion criteria for selected articles.

Identified Articles	Selection	Eligibility	Inclusion
**242** articles	**128** articles	**92** articles	**47** articles
Clinical trials that revealed the beneficial renal or cardiovascular effects.	Hemodynamic effects (renal, cardiovascular, effects on blood pressure, and effects on glomerular function).	Full articles and relevant studies.	High-quality articles. Relevant information.

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
