# Peer review of "Hemodynamic Effects of SGLT2 Inhibitors in Patients with and Without Diabetes Mellitus—A Narrative Review"

_healthcare, 2024, doi:10.3390/healthcare12232464_

Round 1
Reviewer 1 Report
Comments and Suggestions for Authors
In a narrative review, there needs to be a form of process to follow. Your overall review lacked detail in method (although these are not explicit for narrative reviews) it would be better to have some tabulated inclusion/exclusion criteria; and under results, a thematic analysis- so a table of your included studies with themes identified would have been a better approach leading to your discussion of narrative synthesis. I felt this was lacking in your studies along with in-depth critical analysis.
A good paper overall but needs a bit more in-depth critical evaluation and better organisation. Please read my attached comments.
An article that may help with narrative reviews:
Wong G, Greenhalgh T, Westhorp G, et al. RAMESES publication standards: meta-narrative reviews. BMC Med. 2013;11:20.

Please see my comments- authors may wish to add more in-depth analysis to improve the overall quality of the narrative review. There are grammar typos and sentences need to be shorter as they consist of much "jargon" which is fine but needs to be better explained as shorter easier to understand sentences.
Author Response
|
Comments 1: [In a narrative review, there needs to be a form of process to follow. Your overall review lacked detail in method (although these are not explicit for narrative reviews) it would be better to have some inclusion/exclusion criteria; and under results, a thematic analysis- so a table of your included studies with themes identified would have been a better approach leading to your discussion of narrative synthesis. I felt this was lacking in your studies along with in depth critical analysis.] |
|
Response 1: Thank you for pointing this out. I agree with this comment. Therefore, I added a table, called TABLE 1, in Materials and Methods, page 2 and 3, line 92.
|
|
Comments 2: [The review can be significantly strengthened by improving its structure and providing a more in-depth critical analysis of the studies cited. By organizing the discussion in a logical flow, cross-referencing valuable data from tables, and addressing knowledge gaps, the review can offer a more robust and cohesive narrative. A clearer focus on the mechanisms of action, comparisons with other drug classes, and clinical outcomes will not only improve the readability and tone of the review but also enhance its clinical relevance. Additionally, highlighting areas for future research will help guide the next steps in understanding the full potential of SGLT2 inhibitors in clinical practice.] |
|
Response 2: Agree. I revised this to emphasize this point. Page 7, line 275-284. My answer is: ‘’Whereas the inclusions of finding from trials like DAPA-CKD and DAPA-HF strengthens the argument for the therapeutic benefits of SGLT2 inhibitors in CKD (chronic kidney disease) and HF (heart failure), some meta-analysis and randomized controlled trials are cited, providing a robust evidence base for clinical effects (Table 2). Those results, where blood pressure reduction and improvements in HF symptoms, represent the main purpose in this section, can be completed with detailed discussions of renal mechanisms, such as the interaction between Na+/H+ exchanger isoform 3 (NHE3) and SGLT2 inhibitors, macula densa signalling, and adenosine receptor-mediated effects. All of them are well presented in the current review, including multiple hemodynamic effects of SGLT2 inhibitors, especially their natriuretic/diuretic effects.’’ Comments 3: [Comparisons with other drug classes, such as loop diuretics, should highlight the unique mechanisms and benefits of SGLT2 inhibitors. Table 1 (line 249) contains valuable data but is currently underutilized—specific entries from the table should be cross-referenced in the text to strengthen key points, particularly in the discussion of clinical outcomes like mortality reduction and improvements in glomerular filtration rate (GFR).]
Response 3: Agree. I revised this to emphasize this point. Page 10, line 353-365. My answer is: ‘’ A relevant comparison between SGLT2 inhibitors and other drugs can be revealed in the next section. An important study showed that the urine osmolarity and glucose excretion increased from the first day after dapagliflozin administration, compared with individual treatment with bumetanide. In the same study, the daily excretion of uric acid and fluid was greater after combined administration of these drugs, than after bumetanide alone [12]. Regarding serum urate levels we can observe an important effect of dapagliflozin, which, used alone, decreases by 36% these levels, in contrast to the use of bumetanide, which increases this serum level by 3% [12]. Another significant study showed the importance of using these SGLT2 inhibitors in combination with diuretic medication. A relevant example refers to protective cardiovascular effects of SGLT2 inhibitors that surpassed even chlorthalidone [17].]’’
Comments 4: [Whilst it is very difficult to separate the Renin-angiotensin system mechanisms there is an overlap and repetition of information on natriuresis, blood pressure reduction, and cardiac preload/afterload effects. Therefore the authors may wish to revise these sections and address these to introduce focus and clarity in the narrative.]
Response 4: Agree. I revised this to emphasize this point. Page 4, line 127-143. My answer: [These drug classes exhibit significant differences in their effects on potassium, uric acid, glucose, renal hemodynamics, and markers of the renin-angiotensin-aldosterone system, possibly due to their distinct sites of action within the tubular system. Studies are conflicting regarding the diuretic effects of SGLT2 inhibitors, some studies have reported that because of their diuretic/natriuretic effects, they lead to a slightly raised renin levels [11, 12]. Our model predicts that the natriuretic and diuretic effect of SGLT2 inhibition is reduced but maintained in normoglycemia compared to diabetes and with renal impairment compared to normal renal function, which is consistent with recent simulations by Layton and Vallon [11]. Furthermore, we hypothesized that decreases in glomerular pressure and SNGFR were almost independent of renal function and greater in diabetes than in normoglycemia. However, compared to intact renal function, the overall GFR decline was less pronounced in cases of renal impairment. This is because hyperfiltration occurs in diabetic nephrons, and the degree of hyperfiltration in the remaining nephrons increases as nephrons are destroyed. Because of both hyperfiltration and hyperglycemia, the remaining nephrons in diabetics with renal impairment are filtering more glucose and Na+ [11].]
Comments 5: [What is the context of the relationship between SGLT2 inhibitors and other diuretic classes such as loop diuretics and thiazides in terms of their uses within different patient populations and those with volume overload? Need to critique this.
Response 5: Agree. I revised this to emphasize this point. Page 8, line 291-330. My answer: [Table 2 shows the main drugs belonging to the class called SGLT2 inhibitors and their main effects on morbidity and mortality, as follows. The primary composite outcome of a sustained GFR of at least 50%, end-stage kidney disease, or death from renal or cardiovascular causes was less likely to occur in participants with chronic kidney disease, with or without type 2 diabetes, who were randomly assigned to receive dapagliflozin than in those who were assigned to receive a placebo [4]. Another study, based on dapagliflozin SGLT2 inhibitor, presented a large group of individuals with or at risk for atherosclerotic cardiovascular disease, which assessed the drug effects on cardiovascular and renal outcomes. Dapagliflozin treatment did not in a higher or lower rate of MACE (major adverse cardiovascular events) than placebo in patients with type 2 diabetes who had or were at risk for atherosclerotic cardiovascular disease; however, it did result in a lower rate of cardiovascular death or hospitalization for heart failure, which is indicative of a lower rate of heart failure hospitalization [39]. EMPEROR study presents the people from the empagliflozin group who were getting the prescribed treatment for heart failure and were less likely to die from cardiovascular causes or be admitted to the hospital due to heart failure than those in the placebo group [37]. Empagliflozin-induce SGLT2 inhibitors reduced the composite risk of cardiovascular death or heart failure hospitalization by 21% in patients with heart failure and a preserved ejection fraction. This was primarily due to a 29% decreased risk of heart failure hospitalization. Across all predefined categories, including patients with or without diabetes, the effects on the incidence of primary outcome events were largely consistent [38]. Another specific drug with beneficial effects on MACE is canagliflozin. Two from 47 selected articles present the particular impact of a sodium-glucose cotransporter 2 inhibitor, called canagliflozin in clinical practice and how this drug lowers blood pressure, body weight, albuminuria, and glycemia in diabetics [40, 41]. Canagliflozin may also reduce the risk of serious cardiovascular and renal complication, and also death [40]. In conclusion, according to preliminary trails, people with type 2 diabetes may benefit from improved renal outcomes while using SGLT2 inhibitors [41]. Ertugliflozin was not worse than a placebo in patients with type 2 diabetes mellitus and atherosclerotic cardiovascular disease, in terms of significant adverse cardiovascular events [42]. Sotagliflozin therapy, whether started prior to or soon after hospital discharge, reduced the overall number of deaths form cardiovascular causes and urgent visits for heart failure in the trail involving patients with diabetes and a recent episode of acute decompensated heart failure compared to a placebo [43,44].]
Comments 6: [Line 71: The past ten years would be better phrased as from 2014 to 2024.]
Response 5: Agree. I revised this to emphasize this point and thank you so much for attention. Page 2, line 83. My answer: [The search included articles published between 2014 and 2024.]
Comments 7: [I feel that lines 71-76, where the authors describe inclusion /exclusion criteria, could be improved by including a table explaining why 47 articles were included. This would give the findings a bit more robustness and lend substance to the overall study. This will complement the flow chart of Figure 1 which can be improved.]
Response 1: Thank you for pointing this out. I agree with this comment. Therefore, I added a table, called TABLE 1, in Materials and Methods, page 2 and 3, line 92.
Comments 8: [The authors have written their discussion by identifying the themes and this is appropriate for a narrative review but it tends to be rather statement-focused and at times ambivalent in their approach (lines 145–148) as opposed to an in depth critical analysis. Scientists tend to use the words may or possible but I think here you do have a scope for improvement. You have much information in Table 1- perhaps cross reference this in your text as its under-utilised here. This is section 3.5 which is a summary and you should include studies from here within your text to improve and expand upon in depth critical analysis of your narrative.] Response 1: Thank you for pointing this out. I agree with this comment. Therefore, I added text at that section.
Comments 9: [I feel the authors need a more in-depth critical analysis of refs 11,12 to indicate study designs, population studies etc may have an impact on diuretic/natriuretic effects?]
Response 9: Thank you for pointing this out. I agree with this comment. Therefore, I added text at that section.
Comments 10: [Each section starts strongly with a discussion of drug mechanisms and a comparison with other drug classes of similar modes of action, but there is just that in-depth critical analysis lacking, which can come from mentioning the different study designs, settings, population ethnicity, and all the factors that can impact patient outcomes.]
Response 10: Thank you for pointing this out. I agree with this comment. Therefore, I did it at page 8, line 291-330.
Comments 11: [The study is written as a narrative review and can be much improved generally by including more in-depth critical analysis of studies cited, and better integration of tabulated information (Table 1) line 249, within the text as appropriately relevant.]
Response 10: Thank you for pointing this out. I agree with this comment. Therefore, I did it at page 8-9, line 291-333.
|
Reviewer 2 Report
Comments and Suggestions for Authors
Although this work describes the hemodynamic effects of SGLT-2 inhibitors quite well, it is ambitiously called a narrative review. An authentic narrative review should explore, among other things, the mechanisms, clinical significance and generality of the subject. At the same time, this review seems to be unbalanced in both scope and coverage. For instance, the review draws too much on a limited number of prominent studies, like DAPA-HF and EMPEROR-Preserved, while the latest studies and opposing views are barely touched upon. The complexity of the topic, for instance the crosstalk of SGLT-2s and other pathways, or a thorough discussion of side effects, is addressed superficially if at all. More questions remain open though, such as the mechanisms of sodium uptake in distal nephron, which also is not discussed in great detail. Any team undertaking narrative review research intend, and should do, much more with the material collected than this study, which is narrow in ambition, enables.
Some of my issues and recommendations for authors
Issue: The article highlights reductions in cardiovascular events and kidney disease progression but doesn’t fully contextualize these findings against standard treatments or in real-world scenarios.
Recommendation: Compare SGLT-2 inhibitors with other treatment options to highlight their unique advantages or limitations.
Issue: The article heavily cites well-known studies like DAPA-HF, EMPEROR-Preserved, and DAPA-CKD, with little inclusion of newer or less-cited data.
Recommendation: Incorporate findings from smaller, emerging studies or meta-analyses published more recently to provide a more comprehensive perspective.
small p should be written in cursive (p)
Please rephrase Na+ to Na+, same goes with K in all of text
Table 1 should be revised, it is huge, the font is huge
Figure 2 is confusing to me; I do not understand what are the benefits for each organ?
Line 22: “This leads results in reduced blood pressure...” - The phrase "leads results" should be corrected to "leads to".
Line 36: "but soon it was observed the beneficial hemodynamic effects" - This phrase is awkward; consider revising to "but soon the beneficial hemodynamic effects were observed."
Line 61: "...leads tothe observed treatment effects." - "tothe" should be corrected to "to the".
Line 76: "...steps of the search process in illustrated in Figure 1." - "in illustrated" should be corrected to "is illustrated".
Author affiliations and contact information (Lines 6–12): Consistent formatting issues, e.g., missing spaces in some email addresses and a period at the end of the correspondence email.
Random spacing issues, e.g., between "well" and "documented" in line 59.
Line 17: "These drugs, which have limited impact on glycaemic management..." – The term "glycaemic" might need consistency with US English ("glycemic") if that is the intended style.
Line 24: "...introduced in the current guidelines for heart failure management and chronic kidney disease management." – Repeating "management" feels redundant; could revise to "guidelines for managing heart failure and chronic kidney disease."
Line 35: "...soon it was observed the beneficial hemodynamic effects..." – Revision to "the beneficial hemodynamic effects were soon observed" might improve clarity.
Line 39: "Two very large trials demonstrated the beneficial effects of dapagliflozin in heart failure and CKD in diabetes mellitus and non-diabetes mellitus patients (3, 4)." – Awkward repetition of "diabetes mellitus"; consider "patients with and without diabetes mellitus."
Line 54: "...a persistent reduction in the estimated GFR of at least 50% was the primary outcome." – "Primary outcome" might benefit from clarification; consider "primary clinical outcome."

Some typos should be improved
Author Response
|
Comments 1: [small p should be written in cursive (p).]
|
|
Response 1: [In this trial, patients receiving dapagliflozin compared with placebo experienced a lower rate of the primary composite outcome which included worsening heart failure (unplanned hospitalization or urgent visit for heart failure) or death from cardiovascular causes (16.3% vs. 21.2%; p <0.001) [3].]
|
|
Comments 2: [Please rephrase Na+ to Na+, same goes with K in all of text.] |
|
Response 2: Agree.
Comments 3: Table 1 should be revised, it is huge, the font is huge
Comments 4: Figure 2 is confusing to me; I do not understand what are the benefits for each organ? Response 4: thank you so much for mention, and I am so sorry for inconvenience; I made a detailed description in the lower part of the image which makes the image more understandable. I will let it here too: Comments 5: Line 22: “This leads results in reduced blood pressure...” - The phrase "leads results" should be corrected to "leads to". Response 5: Thank you for pointing this out. I/We agree with this comment. Comments 6: Line 36: "but soon it was observed the beneficial hemodynamic effects" - This phrase is awkward; consider revising to "but soon the beneficial hemodynamic effects were observed." Response 6: Thank you for pointing this out. I/We agree with this comment. Comments 7: Line 61: "...leads tothe observed treatment effects." - "tothe" should be corrected to "to the". Response 7: Thank you for pointing this out. I/We agree with this comment. Comments 8: Line 76: "...steps of the search process in illustrated in Figure 1." - "in illustrated" should be corrected to "are illustrated". Response 8: Thank you for pointing this out. I agree with this comment. Comments 9: “Author affiliations and contact information (Lines 6–12): Consistent formatting issues, e.g., missing spaces in some email addresses and a period at the end of the correspondence email.” Response 9: Thank you for pointing this out. I/We agree with this comment. Comments 10: “Random spacing issues, e.g., between "well" and "documented" in line 59.” Response 10: Thank you for pointing this out. I/We agree with this comment. I modified it in the text, at page 2, line 71. Comments 11: “Line 17: "These drugs, which have limited impact on glycaemic management..." – The term "glycaemic" might need consistency with US English ("glycemic") if that is the intended style.” Response 11: Thank you for pointing this out. We agree with this comment, but we changed the abstract and this phrase was cut off, in consequence the term “glycaemic” should not be a problem anymore.
Comments 12: Line 24: "...introduced in the current guidelines for heart failure management and chronic kidney disease management." – Repeating "management" feels redundant; could revise to "guidelines for managing heart failure and chronic kidney disease."
Comments 13: Line 35: "...soon it was observed the beneficial hemodynamic effects..." – Revision to "the beneficial hemodynamic effects were soon observed" might improve clarity.
Comments 14: Line 39: "Two very large trials demonstrated the beneficial effects of dapagliflozin in heart failure and CKD in diabetes mellitus and non-diabetes mellitus patients (3, 4)." – Awkward repetition of "diabetes mellitus"; consider "patients with and without diabetes mellitus."
Comments 15: Line 54: "...a persistent reduction in the estimated GFR of at least 50% was the primary outcome." – "Primary outcome" might benefit from clarification; consider "primary clinical outcome."
|
|
4. Response to Comments on the Quality of English Language |
|
Point 1: |
|
Response 1: (in red)
|
|
5. Additional clarifications |
|
[I just want to thank so much at entire Healthcare team.] |
Round 2
Reviewer 2 Report
Comments and Suggestions for Authors
I am happy with the changes made.
Comments on the Quality of English LanguageCan be improved.